# Development and Evaluation of Sex Pheromone Mass Trapping Technology for *Ectropis grisescens*: A Potential Integrated Pest Management Strategy

**DOI:** 10.3390/insects11010015

**Published:** 2019-12-23

**Authors:** Zongxiu Luo, Fida Hussain Magsi, Zhaoqun Li, Xiaoming Cai, Lei Bian, Yan Liu, Zhaojun Xin, Chunli Xiu, Zongmao Chen

**Affiliations:** 1Tea Research Institute, Chinese Academy of Agricultural Science, 9 Meiling South Road, Xihu District, Hangzhou 310013, China; luozongxiu@tricaas.com (Z.L.); fidajanmagsi@yahoo.com (F.H.M.); zqli@tricaas.com (Z.L.); cxm_d@tricaas.com (X.C.); bianlei@tricaas.com (L.B.); liuyan@tricaas.com (Y.L.); xinzhaojun@tricaas.com (Z.X.); xiuchunli@tricaas.com (C.X.); 2Key Laboratory of Tea Biology and Resource Utilization, Ministry of Agriculture, 9 Meiling South Road, Xihu District, Hangzhou 310013, China

**Keywords:** *Ectropis grisescens*, efficacy evaluation, mass trapping, parameter optimization

## Abstract

Since the identification of the *Ectropis grisescens* sex pheromone, no effective control technology based on this pheromone has yet been developed and evaluated. In this study, pheromone proportion and dosage, sustained-release dispensers, and pheromone lure-matched traps were optimized. The mass trapping technology developed with the above optimized parameters was tested in a field trial. The results show that two compounds, (Z,Z,Z)-3,6,9-octadecatriene and (Z,Z)-3,9-cis-6,7-epoxy-octadecadiene, at a ratio of 30:70 and impregnated into rubber septa at 1 mg, were the most attractive to male moths. These compounds provided the best performance when combined with a sticky wing trap. Adult male moth monitoring data showed that there was a lower population density in the trapping plot compared with the control plot, and there was a clear difference during the peak adult occurrence of the first five insect generations in 2017. The effect of mass trapping on the larva population was investigated in 2018; the control efficiency reached 49.27% after trapping of one generation of adults and was further reduced to 67.16% after two successive adult moth generations, compared with the control plot. The results of the present study provide a scientific basis for the establishment of sex pheromone-based integrated pest management strategies.

## 1. Introduction

The tea plant, *Camellia sinensis* (L.) Kuntze, is one of the most important economic crops in China [1]. By 2018, the Chinese tea growing area had reached 3 million hectares (ha) with an annual output of 1.5 million tons. There have been 808 species of insect and mite pests, belonging to 109 families, recorded in tea plantations [2]. The moth *Ectropis grisescens* Warren (Lepidoptera: Geometridae) is one of the most severe defoliating pests of tea and is distributed throughout the majority of tea-growing areas in China. The *E. grisescens* larval stages consume all of the leaf, leaving only the veins, seriously affecting the quality and productivity of the tea crop. Their mimesis and protective coloration make the *E. grisescens* larvae difficult to see, protecting them from predators. In previous decades, chemical pesticides were the only effective methods for controlling this pest. However, the overuse of pesticides has led to environmental pollution and undesirable chemical residues in the tea product [3]. Currently, semiochemicals are not broadly used in tea pest management; however, the use of sex pheromone compounds for pest management is becoming increasingly popular, and they are likely to play a major role in the future bio-control of tea pests.

Sex pheromones are ideal compounds for insect control technologies because of their specificity and minimal impact on the ecosystem. Synthetic sex pheromones are used for population monitoring, mating disruption, and mass trapping of a wide variety of pest species, particularly members of Lepidoptera [4,5,6]. Although the sex pheromones of some tea pests have been identified, they have not been extensively analyzed, and their use in control strategies remains limited. Only a few studies have been published into the application of sex pheromones in tea pest management. The use of sex pheromones as mating-disruption tools against *Ascotis selenaria cretacea* (Butler) was studied in a Japanese tea plantation. Using a blend of racemic sex pheromones, (*Z,Z*)-6,9-*cis*-3,4-epoxynonadecadiene and (*Z,Z,Z*)-3,6,9-nonadecatriene and their positional isomers, *A. selenaria* mating was successfully disrupted [7]. In another study, different parameters were optimized to suppress the *Euproctis pseudoconspera* (Strand) population using mass trapping with the sex pheromone component (*R*)-10,14-dimethylpentadecyl isobutyrate (*R*-10Me14Me-15:iBu), and the results showed that mass trapping minimized not only the moth population but also the larva and egg densities of *E. pseudoconspera* [8]. Mass trapping is considered a simple and convenient method for the control of *E. pseudoconspera* by smallholders because of its low cost, ease of use, and low labor-intensity.

Previously, both our group and another research team completed the analysis of the sex pheromone of *E. grisescens* [9,10]. The two published studies show the sex pheromone components of *E. grisescens* females to be (Z,Z,Z)-3,6,9-octadecatriene (Z3,Z6,Z9-18:H) and (Z,Z)-3,9-cis-6,7-epoxy-octadecadiene (Z3,epo6,Z9-18:H). However, the reported optimum ratio for field trapping differed from the results obtained during preliminary field testing. Many parameters will need to be optimized before the sex pheromone technology can be applied to pest control; therefore, a series of field tests were designed. The purpose of the present study was to investigate the following factors: (1) the optimum sex pheromone ratio, dosage, sustained-release dispenser, and trap type for mass trapping and (2) the efficacy of mass trapping to control *E. grisescens* adult moth and larval populations in the field. The study aimed to provide sufficient information to develop effective mass trapping technologies based on sex pheromones to control *E. grisescens*.

## 2. Materials and Methods

### 2.1. Chemicals and Field Site

The Z3,Z6,Z9-18:H and Z3,epo6,Z9-18:H used in this study were synthesized by ZeQuan Bio-technology Co., Ltd. (Hangzhou, China), following the method described by Ando et al. [11]. Two synthesized compounds were confirmed to be purer than 97% by gas chromatography–mass spectrometry (Figure 1). All field trials were conducted in a tea plantation of Yuchacun Tea Co., Ltd., in Zhejiang Province, China (29.56° N, 120.41° E) from 2016 to 2018.

### 2.2. Optimization of Sex Pheromone Mass Trapping Technology

#### 2.2.1. Pheromone Ratio Efficiency Experiment

The synthetic sex pheromone compounds were dissolved at 10 µg/µL in hexane. Based on the previous results, Z3,Z6,Z9-18:H and Z3,epo6,Z9-18:H solutions were added to rubber septa (white rubber, 8 mm O.D., Sigma-Aldrich Inc., St. Louis, MO, USA) at ratios of 60:40, 55:45, 50:50, 45:55, 40:60, 35:65, 30:70, 25:75, and 20:80 to provide 1 mg dosage. The prepared lure was combined with the sticky wing trap in the tea plantation. Three replicates of each treatment were tested, and the captured males were counted weekly.

#### 2.2.2. Pheromone Dosage Efficiency Experiment

Four dosages were tested to determine the effect of sex pheromone dosage on male catch numbers. The rubber septa were impregnated with Z3,Z6,Z9-18:H and Z3,epo6,Z9-18:H at 30:70 and 0.1, 0.4, 0.7, 1 mg/septum. The prepared lure was combined with the sticky wing trap in the tea plantation. Six replicates of each treatment were tested, and the captured males were counted weekly.

#### 2.2.3. Pheromone Dispenser Efficiency Experiments

Three kinds of sustained-release dispensers were compared. For isoprene septa and silicone septa dispensers, 10 µg/µL Z3,Z6,Z9-18:H and Z3,epo6,Z9-18:H solutions were directly added at 30:70 and 1 mg/septum. For PVC capillary tubing, Z3,Z6,Z9-18:H and Z3,epo6, Z9-18:H were mixed with corn oil and injected into the PVC capillary tubing. Then, each tube was heat-sealed at 8 cm length (equal to 1 mg total dosage). The prepared lure was combined with the sticky wing trap. Each treatment was replicated four times and randomly set in the tea plantation at 20 m intervals. Captured moths were counted, and sticky boards were replaced every week.

#### 2.2.4. Pheromone Trap Efficiency Experiments

Four different types of trap were tested for the trapping of male *E. grisescens* moths. The traps were as follows: (1) bucket funnel trap (diameter × height of 16 × 20 cm, Pherobio Technology Co. Ltd., Beijing, China); (2) sticky wing trap (length × width × height of 27 × 21 × 14 cm, Enjoy Technology Co, Ltd., Zhangzhou, China); (3) delta trap (length × width × height of 25 × 18 × 16 cm, Enjoy Technology Co, Ltd., Zhangzhou, China); and (4) noctuid trap (diameter × height of 11 × 25 cm, Pherobio Technology Co. Ltd., Beijing, China). All traps were equipped with a rubber dispenser blend of Z3,Z6,Z9-18:H and Z3,epo6,Z9-18:H at 30:70 and 1 mg/septum. Three replicates of each treatment were tested, and the captured males were counted weekly.

### 2.3. Field Efficacy of Sex Pheromone Mass Trapping

#### 2.3.1. Investigation of Population Density of Adult Males

To evaluate the long-term effect of sex pheromone mass trapping on the population density of male *E. grisescens* moths, a large-plot field test was carried out in 2017. Approximately 3 ha of organic tea garden were selected for the mass trapping treatment plot, and sticky wing traps equipped with a rubber dispenser blend of Z3,Z6,Z9-18:H and Z3,epo6,Z9-18:H at 30:70 and 1 mg/septum were arranged on the plot at a density of 60 traps/ha. An organic tea garden of approximately 0.3 ha, 100 m south of the trapping plot, was selected as the blank control. Eight trap devices were arranged in the center of both the treatment and blank control areas to monitor the male moth population. Each of the eight monitoring traps in the treatment plot and blank control plot were investigated every 7 days. The sticky boards in the traps were changed weekly, and the sex pheromone lures were replaced every 2 months. The experiment was conducted from 25 April to 31 October 2017.

#### 2.3.2. Estimation of Larva Control Efficiency

An experiment was carried out in 2018 to determine the control efficiency of mass trapping on the population density of *E. grisescens* larva. In this experiment, 2 ha of organic tea garden were selected for the mass trapping treatment plot, 0.3 ha of organic tea garden situated 100 m from the trapping plot was selected as the blank control, and the trap distribution was as in 2017. The treatment and blank control plots were each divided into four replicates. The initial larva density was investigated before setting the traps. We selected two *E. grisescens* damage centers for each replicate and counted the number of larvae found 1 m^2^ from the damage center. Then, the entire treatment plot was arranged with sex pheromone traps at a density of 60 traps/ha. The sticky boards were replaced every 7 days. The larva population densities of the first and second generations after mass trapping were investigated in June and July. During the investigation periods, no pesticide or fertilizer was used in either the mass trapping treatment or blank control plots, and the agricultural operations were the same in both fields.

### 2.4. Data Analyses

SPSS Statistics 21.0 (IBM, Chicago, IL, USA) was used to analysis the data. The trap-catch data for pheromone ratio, dosage, dispensers, and the trap experiment were log10(*x* + 0.5) transformed to ensure normal distribution and homogeneity before using one-way ANOVA and Tukey’s honestly significant difference (HSD) test at a significance level of *p* < 0.05(“*x*” is the number of captured moth of every trap) The differences in the adult male and larva population densities between mass trapping treatments and the blank control were compared by Student’s *t*-test. The control efficiency of the large-plot mass trapping for larva was calculated using the following formula [12]:$$\mathrm{Control}\;\mathrm{rate}=\left(1-\frac{N_{c\left(x\right)}\times N_{t\left(y\right)}}{N_{c\left(y\right)}\times N_{t\left(x\right)}}\right)\times 100$$
where *N_c(x)_* is the number of larva in the control plot before the start of the experiment, *N_t(y)_* is the number of larva in the treatment plot after mass trapping, *N_c(y)_* is the number of larva in the control plot after mass trapping, and *N_t(x)_* is the number of larva in the treatment plot before mass trapping.

## 3. Results

### 3.1. Pheromone Ratio

Sex pheromone ratios produced different results in the field experiments (Figure 2). Male moth catch sizes varied among the treatments: 60:40 (129.33 ± 12.81 males caught), 55:45 (127.33 ± 17.29), 50:50 (177.33 ± 4.91), 45:55 (140.00 ± 22.90), 40:60 (141.33 ± 20.67), 35:65 (169.33 ± 10.27), 30:70 (201.67 ± 32.73), 25:75 (128.33 ± 20.21), 20:80 (170.33 ± 27.14), and control (5.67 ± 2.03). All capture numbers were higher in the pheromone traps than in the control traps (F = 35.491, df = 9, 20, *p* < 0.01). The sex pheromone binary blend (Z3,Z6,Z9-18:H and Z3,epo6,Z9-18:H) ratio of 30:70 produced the highest trap catch number and was considered the optimal ratio for trapping *E. grisescens*.

### 3.2. Pheromone Dosage

In the field trials, increasing the dosage from 0.1, 0.4, 0.7, to 1 mg/septum increased the size of male moth catches (Figure 3). However, during the test periods, there was no significant difference in trap captures observed among 0.4, 0.7, and 1 mg dosage treatments. There were few males caught at 0.1 mg/septum (137.17 ± 28.37), and there was significantly more caught at 0.4 (261.00 ± 21.84), 0.7 (312.67 ± 19.38), and 1 mg/septum (316.17 ± 19.36) (F = 683.648, df = 4, 25, *p* < 0.01). Furthermore, the dosage optimization field tests revealed that 0.7 mg/septum was almost as attractive as the highest dose of 1 mg/septum. These findings indicate that the optimal dose for mass trapping of *E. grisescens* was between 0.7 and 1 mg/septum.

### 3.3. Pheromone Dispensers

Differences between dispensers (Figure 4) were determined by assessing trap catches; the highest number of moths were caught with isoprene septa (115.75 ± 6.57), followed by silicone septa (108.25 ± 6.09), and PVC capillary tubing (97.25 ± 4.66). However, ANOVA analysis showed no significant differences between the three dispensers (F = 2.711, df = 2, 9, *p* = 0.12), although, isoprene septa performed better in field trials during the entire experimental period compared with silicone and PVC capillary tubing (Figure 5).

### 3.4. Pheromone Traps

Trap efficiency was compared using the number of males caught, and field tests showed there were significant differences among the four types of trap (Figure 6). The sticky wing trap caught significantly more moths (177.00 ± 7.64) than the bucket funnel trap (50.33 ± 10.68), delta trap (34.00 ± 2.89), and noctuid trap (4.00 ± 1.15) (F = 71.504, df = 3, 8, *p* < 0.01). The number of moths caught in the noctuid traps was very low compared with in the bucket funnel and delta traps (Figure 7).

### 3.5. Field Efficacy of Larger Plot Mass Trapping on Population Density of Adult Males

The efficiency of mass trapping for reducing the male population in large plots (Figure 8) was evident by observing the difference in the number of adult males trapped in the treated versus control plots. The *E. grisescens* adult male population density was evaluated from April to October, and one insect generation appeared every month during the experimental period. As Figure 9 shows, there were 15 investigations showing that the population density of the treated plot was significantly lower than that of the control plot. In particular, the population of male moths was significantly suppressed by mass trapping during the peak adult occurrence in the first five generations, compared with in the control plot (25 April *t* = −6.322, df = 14, 9.115, *p* < 0.01; 23 May *t* = −6.373, df = 14, 10.238, *p* < 0.01; 13 June *t* = −5.160, df = 14, 9.499, *p* < 0.01; 10 July *t* = −4.304, df = 14, 13.994, *p* < 0.01; 1 August *t* = −9.769, df = 14, 12.711, *p* < 0.01). In the last generation in October, the population density of both the control and treated plots was low due to the winter diapause.

### 3.6. Efficiency of Large-Plot Mass Trapping for Larva Density Control

Two hectares of organic tea crop were evaluated for the effect of mass trapping on the larval population. Larva numbers in both treated and control plots were investigated before deploying mass trapping (Figure 10), and there was no significant difference between the mean larva population in the control (68.7 ± 58.29) and treated plots (60.23 ± 8.12) (*t* = −0.700, df = 14, 13.994, *p* = 0.495). After trapping one generation of male moths, the population of *E. grisescens* larva illustrated that the control efficiency was 49.27%. The mean number of larvae per square meter of the control plot was 32.38 ± 3.45, which was determined to be significantly higher than in the treated area (13.00 ± 1.94) by ANOVA (*t* = −4.899, df = 14, 11.016, *p* < 0.01). To determine the long-term efficiency of mass trapping, the *E. grisescens* larval population was assessed after two generations; the number of larva per square meter of the control plot (50.38 ± 9.82) was significantly higher than the treated plot (11.63 ± 0.63), according to ANOVA (*t* = −3.939, df = 14, 7.057, *p* < 0.01). The control efficiency increased to 67.16%, demonstrating that mass trapping of *E. grisescens* had a substantial impact on the larva population.

## 4. Discussion

The massive amounts of traditional pesticide used during tea cultivation to control *E. grisescens* have become a major threat to both the tea industry and the ecosystem. Mass trapping and mating disruption are two common integrated pest management (IPM) control strategies that use different mechanisms [13]. Which strategy is chosen depends on the target pest species and its sex pheromone components. Moths of the family Geometridae usually produce type II sex pheromones, i.e., C_17-23_ polyunsaturated hydrocarbons and their epoxy derivatives [14]. However, *E. grisescens* uses two chemical components for chemical communication Z3,Z6,Z9-18:H and Z3,epo6,Z9-18:H [9,15]. In this study, we preferred mass trapping rather than mating disruption to control this pest because mating disruption requires a large amount of pheromone [16], and the synthesis of a uniquely structured sex pheromone or the purification of isomers is a costly process [7,17]. Until development of a low cost synthesis procedure for sex pheromone compounds of *E. grisescens*, mass trapping is an efficient and feasible control method for this pest.

Mass trapping requires the use of highly efficient pheromone traps, ratios, dosages, and dispenser types to suppress the pest population before mating can occur [18,19]. In this study, the efficiencies of different ratios, dosages, dispensers, and traps were assessed. For the sex pheromone components, Z3,Z6,Z9-18:H and Z3,epo6,Z9-18:H, a ratio of 20:80 was previously found to promote the strongest attraction [9], which is similar to the results of our study. However, we tested a greater number of treatment ratios, and the 30:70 treatment ratio was more attractive than 20:80, although there was no difference. With regard to trap efficiency, [20] also optimized the sex pheromone dosage, trap height, and trap types; however, the findings for the pheromone dosage and trap types differ from those of our study. The previous study found bucket funnel traps to be the most effective trap type with a 0.8 mg pheromone dosage, whereas our study found sticky wing traps to be the most efficient, with larger catches (177.00 ± 7.64) than for the bucket funnel traps (50.33 ± 10.68) impregnated with 1 mg *E. grisescens* binary blend sex pheromones. We speculate that successful catching efficiency of the sticky wing trap has two aspects, the design of the trap and zigzag flying pattern of *Ectropis grisescens*. In the context of the flying pattern, the design of the sticky wing trap allows male moths to enter from all of the sides to touch the lure and be easily trapped on sticky boards. Moreover, when we evaluated the pheromone dispensers, we found no significant difference amongst the three kinds of sustained-release dispensers, as found in previous studies [21,22].

Mass trapping techniques have been used to successively suppress the populations of various Lepidoptera pest species [8,23,24,25]. This is the first study to investigate the field efficacy of mass trapping using sex pheromones for the reduction of both adult and larval male *E. grisescens* populations. The results show that mass trapping suppressed peak adult occurrences, when the population is at its highest level. In addition, the overwintered generation, which begins to emerge in early April, was lower than the populations emerging in July and August. Hence, trapping of the overwintered adult population successfully reduced the initial population, which may be a more effective method of mass trapping at the density of 60 traps/ha. Moreover, mass trapping was not only able to control the male population but also to efficiently reduce the larva population of subsequent generations, resulting in less damage to the crop and better quality produce [26,27]. In our study, the population of *E. grisescens* larva was reduced by 49.27% and 67.16% after the trapping of one and two male moth generations, respectively, which is comparable to the control efficiency of biopesticides. We speculated that the number of valid mating candidates for receptive females was dramatically reduced by mass trapping, resulting in a reduced mating success rate.

## 5. Conclusions

*E. grisescens* Warren (Lepidoptera: Geometridae) is most severe pest in tea plantations of China. Different sex pheromone ratio, dosage, and various types of dispensers and traps were evaluated for development of mass trapping technology to control this pest. In conclusion, the present research findings suggest that the sticky wing trap baited with 1 mg binary blend of Z3,Z6,Z9-18:H and Z3,epo6,Z9-18:H impregnated into rubber septa in a ratio of 30:70 provided the most consistent captures of *E. grisescens*. The application of pheromone technology notably reduced the moth population density during all peak occurrences of the male moth and was found to be effective against the larval population. Moreover, the initial population of in April is lower than other generations, possibly because of unfavorable weather conditions for pupae to unsuccessfully emerge. Therefore, application of mass trapping before emergence of overwintered population and continuous application of mass trapping to subsequent generations will be more effective to suppress the pest population. This sex pheromone-based mass trapping technology is an environmentally friendly and efficient method to control *E. grisescens*, and could reduce the application amount of traditional pesticide in tea plantation.

## Figures and Tables

**Figure 1 insects-11-00015-f001:**
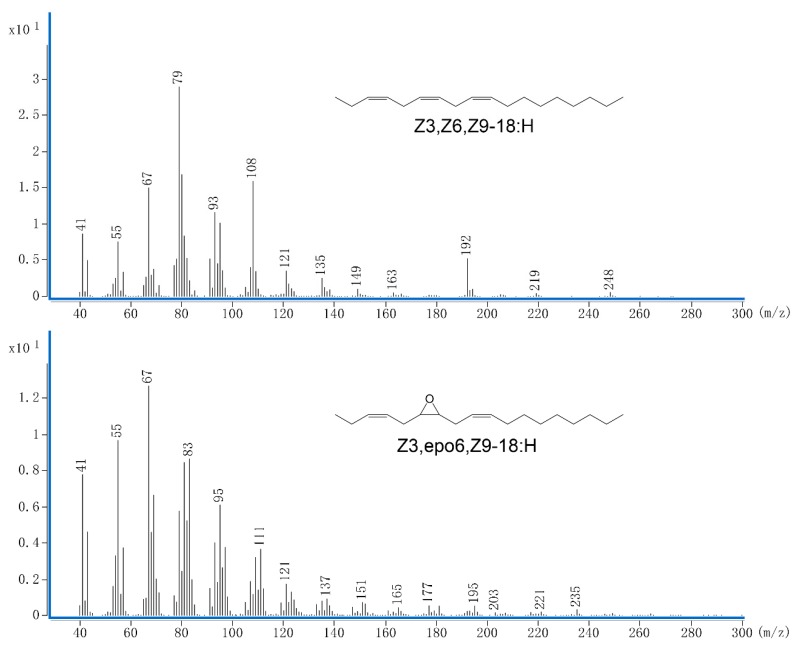
Mass spectrum and chemical structure of two sex pheromone compounds of *Ectropis grisescens.*

**Figure 2 insects-11-00015-f002:**
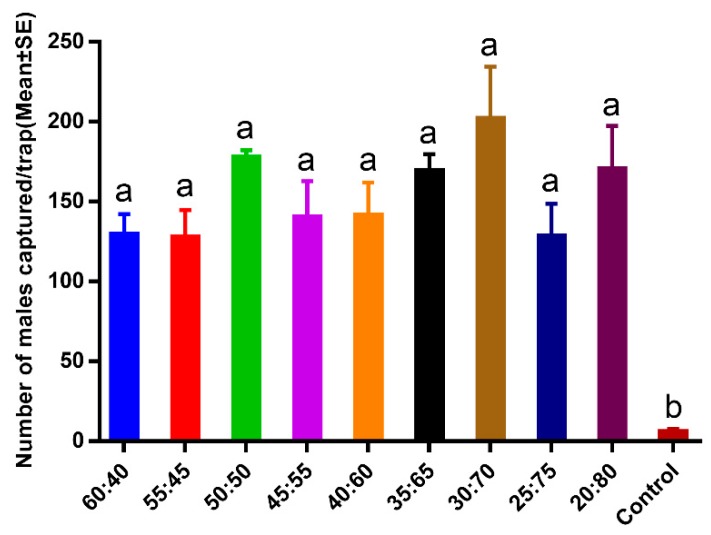
Trap catches of *Ectropis grisescens* male moths with different sex pheromone ratios. Bars mean standard error. Different letters indicate significant difference (Tukey’s honestly significant difference (HSD) test, *p* < 0.05).

**Figure 3 insects-11-00015-f003:**
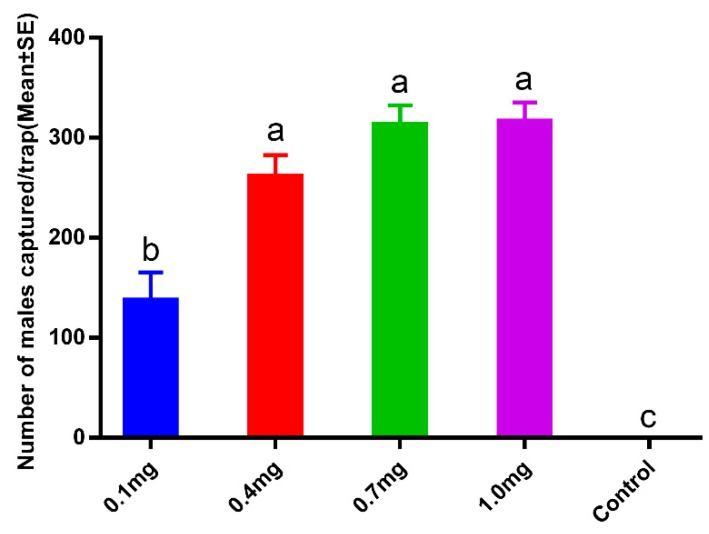
Trap catches of *Ectropis grisescens* male moths with different sex pheromone dosage. Bars mean standard error. Different letters indicate significant difference (Tukey’s HSD test, *p* < 0.05).

**Figure 4 insects-11-00015-f004:**
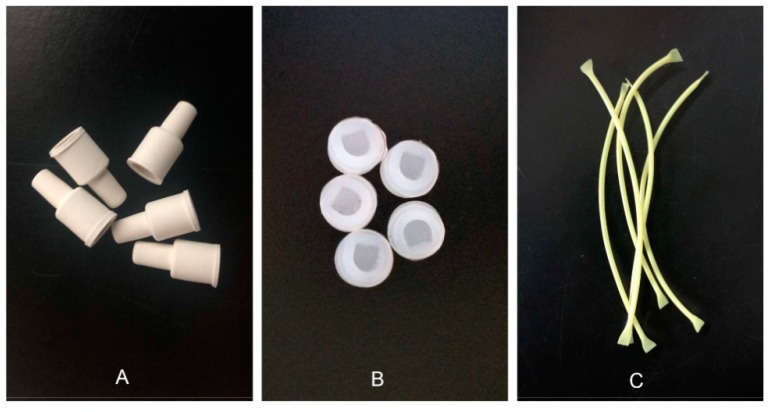
Three kinds of sustained-release dispensers were tested in the field trapping. (**A**) isoprene septa; (**B**) silicone septa; (**C**) PVC capillary tubing.

**Figure 5 insects-11-00015-f005:**
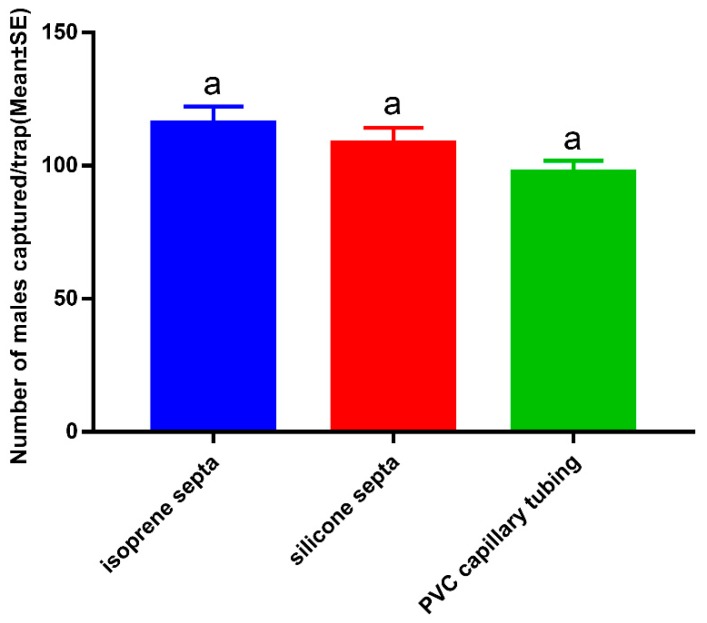
Effect of sex pheromone dispensers on catch of *Ectropis grisescens* male moths. Bars mean standard error. Different letters indicate significant difference (Tukey’s HSD test, *p* < 0.05).

**Figure 6 insects-11-00015-f006:**
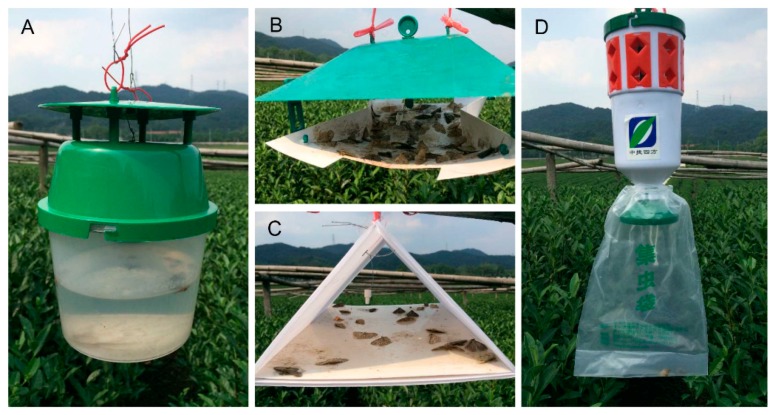
Four kinds of traps were tested in the field trapping. (**A**) bucket funnel trap; (**B**) sticky wing trap; (**C**) delta trap; (**D**) noctuid trap.

**Figure 7 insects-11-00015-f007:**
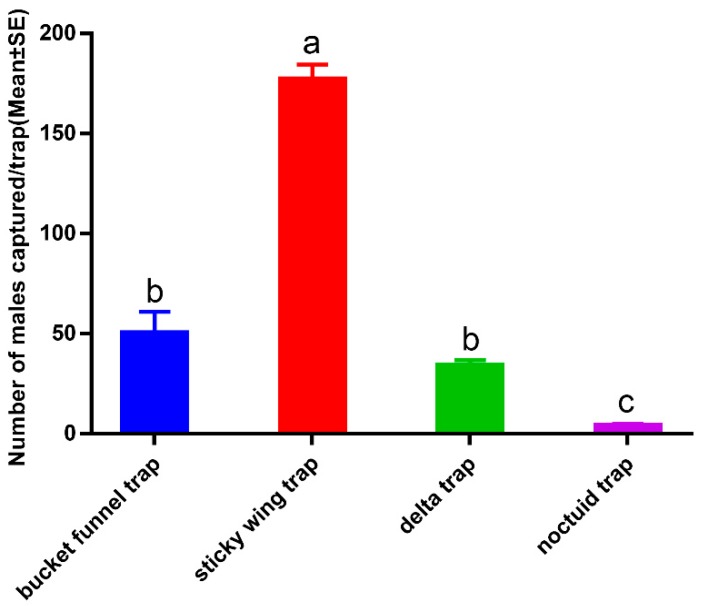
Effect of trap design on catch of *Ectropis grisescens* male moths. Bars mean standard error. Different letters indicate significant difference (Tukey’s HSD test, *p* < 0.05).

**Figure 8 insects-11-00015-f008:**
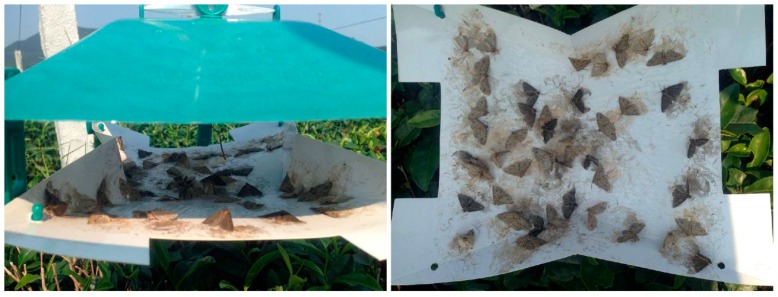
Mass trapping performance of sex pheromone lure of *Ectropis grisescens*.

**Figure 9 insects-11-00015-f009:**
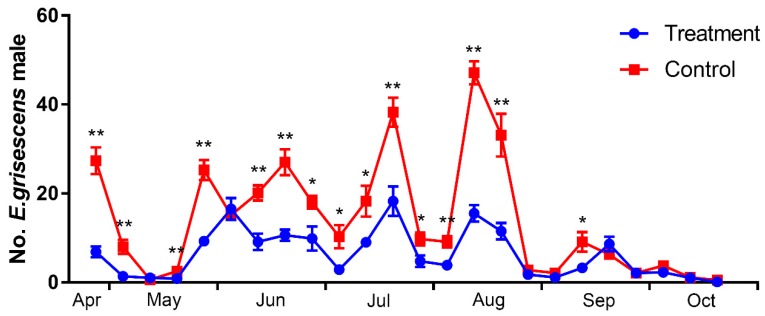
Assessment of larger plot sex pheromone mass trapping trials on population density of *Ectropis grisescens* adult male (25 April–31 October 2017). Difference between mass trapping and control plot was analyzed by Student’s *t*-test, * and ** indicating significant statistical differences between the two species at at *p* < 0.05 and *p* < 0.01, respectively.

**Figure 10 insects-11-00015-f010:**
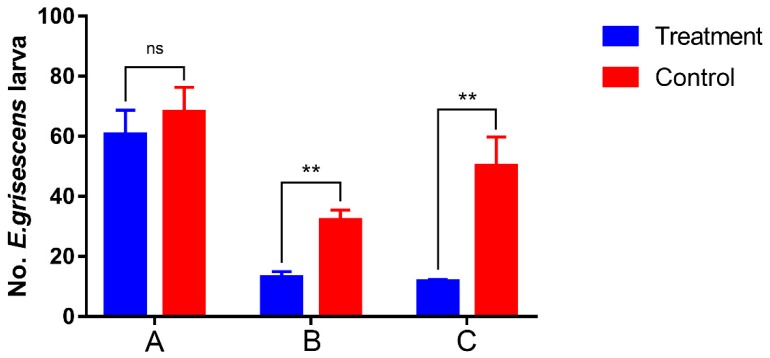
Assessment of larger plot sex pheromone mass trapping trials on population density of *Ectropis grisescens* larva. (**A**) The number of larva in treated and control plots before mass trapping; (**B**) the number of larva in treated and control plots after trapping one generation of male moths; (**C**) the number of larva in treated and control plots after trapping two generations of male moths. Different between mass trapping and control plot were analyzed by Student’s *t*-test, ns means not significant, ** indicating significant statistical differences between the two species at *p* < 0.01.

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
