# Peer review of "Development and Evaluation of Sex Pheromone Mass Trapping Technology for Ectropis grisescens: A Potential Integrated Pest Management Strategy"

_insects, 2019, doi:10.3390/insects11010015_

Round 1
Reviewer 1 Report
Development in the area of crop protection chemicals is very much needed, especially pheromone technology of particular interest. Authors have successfully developed a methodology for effectively utilizing two compounds of pheromone class to trap male moths. I hope this new study definitely increase the use of pheromone technology for better pest management strategies. In this study authors have mainly discussed about different combination ratios of two sex pheromones to make a composition for better pest management. The results are coherent, I recommend publication of the manuscript in the journal Insects.
Review comments for the authors:(have expanded)
For the reader’s point of view, authors must present the chemical structures (Chem-Draw) of the pheromone compounds they used in the manuscript. Authors must present the pictures of different traps and lures they used in the manuscript. Even though descriptive presentation of the results (statistics) with graphs and percentages (line graphs and bar graphs) is necessary, it is recommended that authors should show the photographs /pictures of the different traps/lures at different time intervals they sampled while presenting the results. Results they presented are attractive to read but one cannot believe these results unless they were supported by experimental evidences in the form of pictures/ photographs at each stage of the samples they collected as a supporting information material. From this study one can readily understand the optimum conditions for the best mass trapping of male moths at different conditions but study should be more believable to implement. Finally, manuscript should be published once authors reconstruct the manuscript with concrete evidence for the results of each stage of their study with more pictures / photographs they have sampled at each stage of the study they have presented in the graphs. Recommended to publish after a revision to the current version of the manuscript.Author Response
Please see the attachment

Reviewer 2 Report
The manuscript gives a good contribution to Ectropis grisescens using sex pheromone. However, I have concerns about the discussion and conclusion of the paper that I believe need to be addressed in order to improve its clarity. Their approach is interesting but it has some flaws that make this version unacceptable for publication. Provided they conduct changes to the manuscript, I believe this paper could be of interest to the interested reader on pest control.
L.18: 1 mg
L.23: Delete “,” after adults
L.27: Keywords should be in alphabetic order
Ls.73-74: Please, provide more information about two compounds confirmation by GC/MS method
L.78: µL
L.81: 1 mg
Ls.81-82: How were the males captured? Did you use traps? Explain
L.86: 1 mg
Ls.86-87: Again, How were the males captured?
L.90: 10 μg/μL
L.91: ...and 1 mg/septum
L.93: …captured moths were counted…More information is needed.
L.102: …1 mg/septum
L.109: ..1 mg/septum
L.112-113: … traps were investigated? Confuse, rewrite this sentence.
L.136: Provide references on the used formula in other studies
Ls.142-143: Delete this sentence
L.143: Sentence starting “In the field, Sex pheromone ratios were different…”
L.144: Which moth catch sizes varied among the treatments? Higher or lower catches?
L.147: Delete “significantly”
Ls.148, 177, 190: Delete “Tukey’s HSD test,”
Ls.148, 163, 177, 190, 204, 205, 206, 217, 224: In degrees of freedom, include the two values (treatment and error source)
Ls.150, 165, 199: “E. grisescens” in italic
Ls.156-158: Delete this sentence
Figures 1,2, and 5: Change “Ck” by “Control”
Ls.171-173: Delete this sentence
L.173: Sentence starting “The highest number of moths…”
Ls.185-187: Delete this sentence
Ls.236-238: Sentence repetitive in the introduction section, delete
Ls.238-239: The objective should be in the introduction section. Delete this sentence
L.242: Sentence starting “Species Geometridae produce…”
Ls.245-246: Sentence is unclear, rewrite
L.254: … although there was no different
L.258: …with larger catches
L.259: 1 mg
L.268: Delete “significantly”
Ls.279-287: Paragraph is vague. You should include phrases that provide the most important conclusions of this study (Please, check Ls. 261-265, this is part of your conclusions). Rewrite.
Reviewer 3 Report
This manuscript evaluates a mass-trapping technology against Ectropis grisescens in China. Overall, the manuscript is clearly written,with a focused introduction, a thorough description of the methods and a clear explanation of the results. I only have minor comments. L49, L52: please add the author’s names after the scientific names
L136: Has the formula been used for evaluating the control rate in previous work? In that case, please, add reference ctation(s)
L137-138 and elsewhere: I suggest changing "larva number" with "number of larvae"
L150, L156: please, italicize E. grisescens.
L163 and elsewhere: please remove the decimals for the df values.
Figs 1,2,3,4: I suggest changing the y-axis title with Number of males captured/trap (mean ± SE)"
L204-206: Rather than comparing the captures at each timing, you may also think to fit non linear models to the cumulative catches
per each flight period, then compare treatment vs. Ck (e.g. https://doi.org/10.1046/j.1439-0418.2001.00594.x).
L257: The higher catches by the sticky wing trap (Fig. 4) is interesting and should be discussed more in details
Round 2
Reviewer 2 Report
The manuscript “Development and evaluation of sex pheromone mass trapping technology
for Ectropis grisescens: a potential integrated pest management strategy” has been improved and all my questions were taken into account.
I recommend the publication in “Insects”.
Author Response
Thank you for your professional review work on our article. It is our honor to publish our manuscript on ‘insects’ after corrected some mistakes according your valuable suggestion.